# Metabolite Profiling and Bioassay-Guided Fractionation of *Zataria multiflora* Boiss. Hydroethanolic Leaf Extracts for Identification of Broad-Spectrum Pre and Postharvest Antifungal Agents

**DOI:** 10.3390/molecules27248903

**Published:** 2022-12-14

**Authors:** Ali Karimi, Torsten Meiners, Christoph Böttcher

**Affiliations:** 1Institute for Ecological Chemistry, Plant Analysis and Stored Product Protection, Julius Kuehn Institute, Koenigin-Luise-Straße 19, 14195 Berlin, Germany; 2Institute of Pharmacy, Freie Universitaet Berlin, Koenigin-Luise-Straße 2-4, 14195 Berlin, Germany

**Keywords:** bioassay-guided fractionation, flavonoids, hydroxycinnamic acid derivatives, Lamiaceae, metabolite profiling, phytopathogenic fungi, terpenoids, *Zataria multiflora* Boiss.

## Abstract

Hydroethanolic leaf extracts of 14 Iranian *Zataria multiflora* Boiss. populations were screened for their antifungal activity against five plant pathogenic fungi and metabolically profiled using a non-targeted workflow based on UHPLC/ESI-QTOFMS. Detailed tandem mass-spectrometric analyses of one of the most active hydroethanolic leaf extracts led to the annotation of 68 non-volatile semi-polar secondary metabolites, including 33 flavonoids, 9 hydroxycinnamic acid derivatives, 14 terpenoids, and 12 other metabolites. Rank correlation analyses using the abundances of the annotated metabolites in crude leaf extracts and their antifungal activity revealed four *O*-methylated flavones, two flavanones, two dihydroflavonols, five thymohydroquinone glycoconjugates, and five putative phenolic diterpenoids as putative antifungal metabolites. After bioassay-guided fractionation, a number of mono-, di- and tri-*O*-methylated flavones, as well as three of unidentified phenolic diterpenoids, were found in the most active subfractions. These metabolites are promising candidates for the development of new natural fungicides for the protection of agro-food crops.

## 1. Introduction

With more than 3000 species, the Nepetoideae is the largest subfamily in the Lamiaceae family [1]. It comprises numerous genera rich in essential oils and includes important culinary and medicinal herbs, such as thyme, oregano, rosemary, sage, savory, basil, mint, and lemon balm. One of the smallest genera in the subfamily Nepetoideae is *Zataria* with only one described species, *Zataria multiflora* Boiss. *Z. multiflora* is a thyme-like, perennial aromatic shrub native to southwest Asia (Afghanistan, Iran, Oman, Pakistan). Similar to thyme, the aerial parts of *Z. multiflora* are used as a spice and in traditional folk remedies for their antiseptic, analgesic, carminative, anthelmintic, and antidiarrheal properties [2]. Its essential oil (EO) consists mainly of oxygenated monoterpenes and monoterpene hydrocarbons with carvacrol, thymol, linalool, *γ*-terpinene, and *p*-cymene as major components [3]. Due to the antibacterial and antifungal properties of the phenolic monoterpenes thymol and carvacrol, the EO of *Z. multiflora* is used in pharmaceutical preparations for the treatment of fungal infections of the skin, coughs, bronchitis, and digestive disorders [2,4].

In addition to volatile monoterpenes, Nepetoideae species produce a plethora of non-volatile secondary metabolites with a broad spectrum of bioactivities [1]. Non-volatile compounds include phenolic diterpenoids (e.g., carnosol, carnosic acid), caffeic acid derivatives (e.g., rosmarinic acid, salvianolic acids), and various flavonoids, such as multiple hydroxylated or methoxylated flavones and flavanones. Despite its regional importance as an aromatic and medicinal plant, the fraction of non-volatile secondary metabolites from the leaves of *Z. multiflora* is less well explored. So far, only a limited number of metabolites have been identified or putatively annotated, including the thymol/carvacrol derivatives zataroside A and B, zatatriol, multiflotriol, multiflorol, the caffeic acid derivatives rosmarinic acid, lithospermic acid A and B, as well as luteolin, luteolin *O*-glucuronide, vicenin-2, and naringenin as flavonoids [5,6,7].

Fungal diseases are the main cause of severe pre- and post-harvest losses in agriculture and horticulture [8]. Among the broad spectrum of fungal pathogens, the genera *Fusarium, Botrytis*, *Alternaria*, and *Colletotrichum* have a high potential to infect and destroy various crop plant species in pre- and post-harvest stages, as well as contaminate them with mycotoxins that can harm human and animal health [9]. For example, the grey mold fungus, *Botrytis cinerea*, causes enormous damage to more than 200 crop species, including small fruit crops and vegetables [10]. Synthetic fungicides have been the silver bullet in the fight against these pathogens for decades. Due to their increasing resistance to existing synthetic fungicides, their environmental toxicity, and stricter regulation, the demand for alternative strategies to control fungal diseases in food crops has increased. One alternative to synthetic fungicides is the use of natural substances that either have antifungal properties or stimulate the plants’ natural defense mechanisms. Medicinal and aromatic plants in particular are a rich source of biologically active natural products that can be used not only to combat human diseases and preserve food, but also to develop novel bio-based pesticides for agricultural purposes [11,12].

In recent studies, we investigated the composition and antifungal activities of EOs from leaves of *Z. multiflora* collected from 14 different natural habitats in Iran [3,13]. The observed chemical diversity of EOs and their different antifungal activities prompted us to further characterize the fraction of non-volatile semi-polar secondary metabolites in this sample set with respect to their composition and antifungal activity. To this end, hydroethanolic leaf extracts of *Z. multiflora* populations were screened for their antifungal potential against five plant pathogenic fungi and classified based on their metabolite profiles obtained by ultra-high-performance liquid chromatography coupled with electrospray ionization quadrupole time-of-flight mass spectrometry (UHPLC/ESI-QTOFMS). After in-depth characterization of the most active leaf extract using tandem mass spectrometric techniques, correlational approaches and bioassay-guided fraction were applied to identify antifungal metabolites as candidate bio-based pesticides.

## 2. Results

### 2.1. Screening of Antifungal Activities of Hydroethanolic Leaf Extracts

Hydroethanolic leaf extracts of *Z. multiflora* from 14 geographically and environmentally diverse natural habitats in Iran were assayed for their inhibitory effect on the mycelial growth of five phytopathogenic fungi (Table 1). In general, the leaf extracts showed low to moderate, and in some cases even high antifungal activities (inhibition rate > 60%). For all five fungal species tested, antifungal activities were significantly different between *Z. multiflora* populations (ANOVA, 2.54 × 10^−15^ ≤ *p_adj_* ≤ 1.8 × 10^−7^). Leaf extracts from Siriz and Haneshk populations had by far the lowest activity against all tested fungi. In contrast, leaf extracts from Daarbast, Hongooyeh, Jandaq, and Konar Siah populations were the most active. Especially in assays with *A. dauci* and *C. lindemuthianum*, the leaf extracts from Jandaq and Daarbast populations caused high mean growth inhibition rates of 64.9% and 65.7%, respectively.

### 2.2. Metabolite Profiling of Hydroethanolic Leaf Extracts

To characterize the metabolite composition, hydroethanolic leaf extracts used for antifungal bioassays (14 populations × 3 samples) were diluted and analyzed by reversed-phased UHPLC/DAD/ESI-QTOFMS in positive and negative ion mode. After feature detection and alignment using the XCMS algorithm, two data sets comprising 12,783 and 10,518 molecular features were obtained from the raw data acquired in positive and negative ion mode, respectively. Both data sets were subjected to unsupervised random forest (RF) classification. For data from positive ion mode, visualization of the proximity matrix by multidimensional scaling (Figure 1) revealed three main clusters. Cluster I comprised samples from the Siriz and Haneshk populations, which had the lowest antifungal activities. Samples of the Fasa, Taft, Arsenjan, and Kemeshk populations were located in cluster II, while cluster III included samples from the Darab, Ashkezar, Hongooyeh, and Gezeh populations. Samples from three of the four populations with the highest antifungal activities, including Jandaq, Konar Siah, and Daarbast, and samples from the Gachooyeh population were scattered in both cluster II and III. A corresponding analysis using data from negative ion mode (Appendix A) generally confirmed the distribution of populations among the three clusters.

To estimate the variation in metabolite levels between populations, abundances of the detected molecular features were analyzed by ANOVA with population as a factor. Significant differences in abundance between populations were found for 56.7% (7248/12,783) of the molecular features detected in positive and for 61.5% (6479/10,518) of those detected in negative ion mode (*p_adj_* ≤ 0.01).

### 2.3. Annotation of Semi-Polar Secondary Metabolites

Due to the high metabolic diversity of the sample set, the in-depth characterization of semi-polar secondary metabolites was limited to a single population that exhibited high antifungal activity and was scattered in clusters II and III. For this purpose, a pooled sample of all leaf extracts from the Konar Siah population was prepared and subjected to accurate mass tandem mass spectral analysis. Based on library searches and manual interpretation of the collision-induced dissociation (CID) mass spectra obtained, a total of 68 metabolites were putatively annotated (Table 2), including 33 flavonoids (1–32), 9 hydroxycinnamic acid (HCA) derivatives (33–41), 14 terpenoids (42–55) and 12 other metabolites (56–67). Of these, 17 were clearly identified using commercially available reference compounds. Representative chromatograms with peak annotation are given in Figure 2, and molecular structures are shown in Appendix A. Full chromatographic and mass spectral data and comments on compound identification and fragmentation schemes are given in Appendix A.

#### 2.3.1. Flavonoids

About half of the annotated metabolites were flavonoids. In general, three classes of flavonoids were detected, including flavones and derived *O*- and *C*-glycoconjugates (1–26), flavanones (27–30), and dihydroflavonols (31–32) (Table 2). Apigenin, luteolin, and 6-hydroxyluteolin were identified as aglycones in a number of glycoconjugates (5–7, 12–14, 15–16). Mono-*O*-glycosides of the three flavones carrying a hexosyl (5, 12, 15) and a hexuronosyl (6, 13, 16) moiety were annotated. In addition, di-*O*-glycosides with a deoxyhexosyl → hexosyl moiety (7, 14) were detectable exclusively with apigenin and luteolin as aglycones. Using reference compounds, two of the *O*-hexosides (5, 12) were identified as apigenin 7-*O*- and luteolin 7-*O*-*β*-d-glucopyranoside. Besides the annotated *O*-glycosides, *C*-glycosidic conjugates of apigenin (2–4) and luteolin (9–11) were found to accumulate in leaves of the Konar Siah population. These are two near-eluting apigenin (2, 3) and luteolin *C*-hexosides (9, 10), which probably differ only in the glycosylation position (6-*C*/8-*C*) as well as corresponding 6,8-di-*C*-hexosides (4, 11). Among the *C*-glycosylated flavones, apigenin 6,8-di-*C*-hexoside (4) was the quantitatively dominating compound. Based on the characteristic fragment ions formed by cross-ring cleavage of the *C*-glycosyl moieties upon CID, metabolite 30 was annotated as di-*C*-hexoside of C_15_H_12_O_6_. The cross-ring fragmentation of deprotonated 30 is superimposed by a prominent neutral loss of 162.032 (C_9_H_6_O_3_) which is characteristic for *C*-glycosylated ring-open 2-hydroxynarigenin derivatives [14].

Besides the above-mentioned unmethylated flavones, a large number of *O*-methylated flavones carrying one to three methyl groups were detected (17–26). Flavones with one methyl group include 7-*O*-methyl-apigenin (genkwanin, 17), 7-*O*-methyl-luteolin (20), 3′-*O*-methyl-luteolin (chrysoeriol, 19-1), and 6-*O*-methyl-scutellarein (6-methoxy-apigenin, hispidulin, 19-2), which were identified using authentic reference compounds. Chrysoeriol and hispidulin co-elute under chromatographic conditions used for metabolite profiling but can be partially separated and authenticated using a modified eluent system (Appendix A). Another mono-*O*-methylated flavone (18) was detected but finally not identified. Based on characteristic fragment ions detected at *m/z* 168.005 ([^1,3^A-CH_3_]^•+^), *m/z* 121.029 (^0,2^B^+^), and *m/z* 119.050 (^1,3^B^+^), it was concluded that ring A of 18 carries two hydroxy and one methoxy group, and accordingly ring C only one hydroxy group. Flavone 18 could thus be a scutellarein derivative that is methylated in 7-*O* or 5-*O* position. While unmethylated flavones accumulate mainly as glycoconjugates, only chrysoeriol and hispidulin have been detected as aglycones in two mono-*O*-glycosides (21, 22). Besides mono-*O*-methylated flavones, three di-*O*-methylated (23–25) and one tri-*O*-methylated flavone (26) have been detected, one of which (24) was putatively annotated as 3’,7-di-*O*-methylluteolin (7-*O*-methyl-chrysoeriol, velutin) based on a reference spectrum published in MassBank. The other higher *O*-methylated flavones are probably di-*O*-methyl scutellarein (23) and di- and tri-*O*-methylated 6-hydroxyluteolin derivatives (25, 26).

In the leaves of *Z. multiflora*, the diversity of flavonoids is even greater due to the presence of three flavanones (27–29) and two dihydroflavonols (31, 32). Naringenin (27), 3´-*O*-hydroxynaringenin (eriodictyol, 28), and an *O*-methylated naringenin derivative (29) were annotated as flavanones, the first two of which were identified using authentic reference compounds. CID of protonated 29 resulted in ^1,3^A^+^ and ^1,4^B^+^ fragment ions detected at *m*/*z* 167.034 and *m*/*z* 147.045, respectively. Compared to naringenin, *m*/*z* of ^1,3^A^+^ derived from protonated 29 was shifted by 14.015 amu, while *m*/*z* of ^1,4^B^+^ remained constant (Appendix A). Thus, flavanol 29 carries its methyl group on ring A and, based on the presence of other 7-*O*-methylated flavonoids, probably represents 7-*O*-methylnaringenin (sakuranetin). Dihydrokaempferol (aromadendrin, 31) and dihydroquercetin (taxifolin, 32) were identified as dihydroflavonols based on reference compounds.

#### 2.3.2. Hydroxycinnamic Acid Derivatives

A total of nine metabolites (33–41) were annotated as HCA derivatives. In the extracted wavelength chromatogram at 260–340 nm of the pooled sample of the Konar Siah population (Figure 2) the highest peak eluting at 5.69 min was assigned to rosmarinic acid (33) using an authentic reference compound. In a retention time range from 5.4 to 5.9 min, rosmarinic acid is accompanied by other prominent metabolites (34–37) that show similar absorption spectra to rosmarinic acid with absorption maxima around 287 and 332 nm. Under negative ion electrospray conditions, metabolites 34–37 formed deprotonated molecular ions at *m*/*z* 717.1461 (34), *m*/*z* 1075.2156 (35), *m*/*z* 1075.2149 (36), and *m*/*z* 1433.2789 (37), while for metabolites 35–37 double deprotonated molecular ions were also detectable at *m*/*z* 537.1038 (35), *m*/*z* 537.1045 (36), and *m*/*z* 716.1388 (37). Therefore, metabolites 34–37 probably represent a series of dehydro-oligomers of rosmarinic acid with elemental compositions of C_18+18*n*_H_16+14*n*_O_8+8*n*_ (*n* = 1–3). Deprotonated 34 showed sequential neutral losses upon CID corresponding to dihydroxyphenyllactic acid (C_9_H_10_O_5_, 198.052 amu) and dehydrated caffeic acid (C_9_H_6_O_3_, 162.032 amu), resulting in a relatively stable fragment ion at *m*/*z* 357.061 (C_18_H_13_O_8_^−^) from which losses of carbon dioxide and water were observed at higher collision energies. The CID mass spectrum of deprotonated 34 is therefore consistent with a dehydrodimer of rosmarinic acid in which two oxidized rosmarinic acid moieties are linked via a C–C bond, probably forming a phenylcoumaran linkage motif as described for salvianolic acid B/lithospermic acid B. As observed for dehydrodimer 34, CID of deprotonated dehydrotrimers 35 and 36 and dehydrotetramer 37 occurred mainly along the ester bonds. The CID mass spectra of deprotonated 35–37 showed the same set of fragment ions at *m*/*z* 555.114, *m*/*z* 519.093, and *m*/*z* 357.061 as observed in the CID mass spectrum of deprotonated 34. In addition, analogous fragment ions were observed to be shifted by a mass increment of 358.069 amu (C_18_H_14_O_8_).

Besides rosmarinic acid and its dehydrooligomers as quantitatively dominant HCA derivatives, minor amounts of caffeic acid (38), chlorogenic acid (39) and coumaric acid 4-*O*-hexoside (41) were also detectable in leaf extracts of the Konar Siah population. Another putative HCA derivative with an elemental composition of C_17_H_14_O_6_ was detected in positive and negative ion modes at *m*/*z* 315.0863 ([M+H]^+^) and *m*/*z* 313.0718 ([M-H]^-^), respectively. CID of protonated 40 revealed one prominent fragment ion at *m*/*z* 163.039 (C_9_H_7_O_3_^+^) and two lower abundant fragment ions at *m*/*z* 205.049 (C_11_H_9_O_4_^+^) and *m*/*z* 161.059 (C_10_H_9_O_2_^+^). At higher collision energies, daughter ions of *m*/*z* 163.039 were formed, characteristic of a caffeoyl moiety. CID of deprotonated 40 induces fragmentation along the ester bond, resulting in two complementary fragment ions at *m*/*z* 161.024 (C_9_H_5_O_3_^−^) and *m*/*z* 151.040 (C_8_H_7_O_3_^−^). Based on elemental composition and fragmentation data from literature [15], metabolite 40 was annotated as 2-(dihydroxyphenyl)ethenyl caffeate, which could represent nepetoidin A or B or geometric isomers thereof, depending on the position of the hydroxy groups.

#### 2.3.3. Terpenoids

A number of metabolites were detected, comprising an unconjugated aglycone (42), two derived *O*-hexosides (43, 44), two *O*-(*O*-malonyl-hexosides) (45, 46), and one *O*-(*O*-hexosyl-hexoside) (47). In contrast to 42, which was ionized exclusively under negative ion electrospray conditions and formed deprotonated molecular ions, glycoconjugates 43–47 were detectable in both ion modes. Upon CID, the ammonium adducts of 43 and 44 formed abundant [B_1_ + NH_3_]^+^ and Y_0_^+^ ions at *m*/*z* 180.086 (C_6_H_14_NO_5_^+^) and *m*/*z* 167.107 (C_10_H_15_O_2_^+^), respectively. In addition, cleavage of propene from Y_0_^+^ was observed, resulting in a fragment ion at *m*/*z* 125.060 (C_7_H_9_O_2_^+^). Deprotonated 43 and 44 formed upon CID ^0,2^X_0_^−^, Y_0_^−^, [Y_0_-H]^•−^ and B_1_^-^ fragment ions at *m*/*z* 207.102 (C_12_H_15_O_3_^−^), *m*/*z* 165.092 (C_10_H_13_O_2_^−^), *m*/*z* 164.084 (C_10_H_12_O_2_^•−^), and *m*/*z* 161.045 (C_6_H_9_O_5_^−^), respectively. At higher collision energies, the formation of a [Y_0_-CH_4_]^-^ fragment ion was observed. Therefore, 43 and 44 were dereplicated as zataroside A/B being *β*-*O*-glucopyranosides of thymohydroquinone (THQ, 2,5-dihydroxy-*p*-cymene) [6]. Based on mass spectral similarities, 45 and 46 were putatively annotated as THQ *O*-(*O*-malonyl-hexosides), 47 as THQ *O*-(*O*-hexosyl-hexoside), and 42 as unconjugated THQ.

Another series of eight metabolites (48–55) was detected in the rear part of the chromatogram between 8.5 and 10.6 min. Metabolites 48–55 consistently formed deprotonated molecular ions under negative ion electrospray conditions. The elemental compositions determined for 48–55 (C_20_H_26_O_3-4_ and C_20_H_24_O_3-5_) indicate a series of diterpenoids with C_20_ skeletons, 8 to 9 ring and double bond equivalents, and 3 to 5 oxygen atoms. The CID mass spectra obtained from the quasi-molecular ions of 48–55 were difficult to interpret and do not allow clear conclusions to be drawn about the underlying molecular structures.

#### 2.3.4. Other Metabolites

Besides flavonoids, HCA conjugates, and terpenoids, 12 other metabolites (56–67) belonging to different biosynthetic classes were annotated. A nitrogen-containing hexoside (56) with the elemental composition of C_15_H_17_NO_6_ was detected in positive and negative ion modes at *m*/*z* 346.0897 ([M + Na]^+^) and *m*/*z* 322.0932 ([M-H]^-^), respectively. CID of deprotonated 56 revealed ^0,4^X_0_^−^, ^0,2^X_0_^−^, Y_0_^−^, [Y_0_-CO_2_]^-^ fragment ions at *m*/*z* 262.072 (C_13_H_12_NO_5_^−^), *m*/*z* 202.050 (C_11_H_8_NO_3_^−^), *m*/*z* 160.040 (C_9_H_6_NO_2_^−^), *m*/*z* 116.050 (C_8_H_6_N^−^), indicating an *O*- or *N*-hexoside of an indole carboxylic acid. CID of the in-source fragment ion Y_0_^+^ detected at *m*/*z* 162.055 (C_9_H_8_NO_2_^+^) produced a similar spectrum to that from authentic protonated indole-3-carboxylic acid. Finally, metabolite 56 exhibited the same chromatographic and mass spectral characteristics as *β*-glucosyl indole-3-carboxylate detected in a hydromethanolic extract of silver nitrate treated *Arabidopsis thaliana* leaves [16]. A series of three metabolites consisting of an unconjugated aglycone (57) with the elemental composition of C_12_H_18_O_4_, a derived *O*-hexoside (58), and *O*-(*O*-malonylhexoside) (59) were detected. CID of protonated 57 showed sequential losses of two molecules of water and acetic acid. Deprotonated 57 produced upon CID almost exclusively a fragment ion at *m*/*z* 59.013 (C_2_H_3_O_2_^−^), which is also prominently formed upon CID of deprotonated jasmonic acid (C_12_H_18_O_3_). Hence, metabolite 57 was putatively annotated as hydroxyjasmonic acid, metabolite 58 and 59 as hydroxyjasmonic acid *O*-hexoside, and *O*-(*O*-malonylhexoside), respectively. As phenylmethanoids, 3,4-dihydroxybenzaldehyde (60), an *O*-hexosylated and *O*-pentosylated dihydroxybenzoic acid (61), an *O*-hexosylated dimethoxyhydroxybenzoic acid (62), and benzylalcohol *O*-(*O*-malonylhexoside) (67) were putatively annotated. Two glycosylated tyrosol derivatives carrying a hexosyl (65) and a pentosyl → hexosyl moiety (66) were detected. Metabolite 65 did not form pseudo-molecular ions under ESI conditions, instead formation of the adduct ions [M + NH_4_]^+^ at *m*/*z* 318.1543 and [M + HCOO]^−^ at *m*/*z* 345.1194 was observed. CID of the formiate adduct of 65 resulted in formation of a C_1_^−^ and a Z_0_^-^ fragment ion at *m*/*z* 179.056 (C_6_H_11_O_6_^−^) and *m*/*z* 119.050 (C_8_H_7_O^-^), suggesting that the hexosyl moiety is linked to tyrosol via its aliphatic hydroxy group. In addition to the indole derivative 56, tyramine and *N*-γ-glutamyl tyramine were annotated as further nitrogen-containing secondary metabolites.

### 2.4. Correlation of Metabolite Abundance with Antifungal Activity

The annotated metabolites were quantified relatively in the complete sample set, which included 42 samples from 14 populations. As shown by ANOVA, the abundances of metabolites 1–67 were significantly different between populations (1.44 × 10^−8^ ≤ *p*_adj_ ≤ 2.72 × 10^−2^). Plotting the mean abundances of the populations in a heat map showed large differences for some metabolites (Figure 3). In particular, the leaves of the Siriz and Haneshk populations from cluster I had comparably low levels of methylated flavones (18, 20, 23, 25, 26), flavanones (27, 28), dihydroflavonols (31, 32), and of terpenoids (42–55). In contrast, the leaves of the Darab, Ashkezar, Hongooyeh, and Gezeh populations from cluster III had higher contents of the putative diterpenoids #1–#8 (48–55). To examine the correlation structure of metabolite abundances, a debiased sparse partial correlation network was constructed (Appendix A), which revealed three subnetworks with at least three nodes and several modules of highly connected metabolites. These modules contained apigenin and luteolin *C*-glycosides (2–4, 9–11), apigenin, luteolin, and 6-hydroxyluteolin *O*-glycosides (5, 12, 15, 16), putative diterpenoids (48–50, 52–55), THQ glycosides (43–47), flavanones (27, 28), dihydroflavonols (31, 32), and methylated flavones (18–23, 25, 26).

To identify candidate metabolites with putative antifungal activity, metabolite abundances of individual samples were rank correlated with corresponding inhibition rates from the initial screening experiment (Figure 3). Significance analyses revealed that 140 out of 335 calculated rank correlations were statistically significant (*p*_adj_ ≤ 0.05). Of these, 94 had a positive correlation and 46 had a negative correlation. Significant positive correlations between abundance and antifungal activity against all five fungi were registered for four methylated flavones, including trihydroxy-methoxy-flavone (18), 7-*O*-methylluteolin (20), trihydroxy-dimethoxy-flavone (25), and dihydroxy-trimethoxy-flavone (26). The levels of naringenin (27), eriodictyol (28), dihydrokaempferol (31), and dihydroquercetin (32) were also significantly positively correlated with antifungal activity in all five fungal species. Among the terpenoids, the THQ glycoconjugates (43–47) showed significant positive correlations with antifungal activity, but not THQ (42) itself. The putative diterpenoids #1–#3 (48–50) and #7–#8 (54–55) also showed significant positive correlations between abundance and antifungal activity against at least four fungal species.

### 2.5. Bioassay-Guided Fractionation of a Hydroethanolic Leaf Extract

To validate the results from correlation analyses, a classical bioassay-guided fractionation of a hydroethanolic extract prepared from a pooled leaf sample of the Konar Siah population was performed. As in the first screening experiment, the same five fungal species were used for monitoring the antifungal activity of the prepared fractions. For a first rough fractionation, the total hydroethanolic extract was separated on Strata C18-E solid phase extraction cartridges using a methanol/water step gradient (Figure 4A). Of the five fractions obtained, fractions 3 and 4, eluted with 60% and 80% aqueous methanol, respectively, showed moderate antifungal activities against all fungal species (Figure 4B). However, for four out of five fungi tested, fraction 4 showed significantly higher antifungal activities compared to fraction 3 (Appendix A). Therefore, fraction 4 was further separated by semi-preparative reversed-phase HPLC, and the resulting seven subfractions were bioassayed (Figure 4B). As shown by UHPLC/DAD/ESI-QTOFMS, subfraction 4-1 to 4-7 contained a number of methylated flavones (17, 19-1/2, 20, 25, 26), apigenin (1), sakuranetin (29), and the putative diterpenoids #1–#3 (48–50). Remarkably, three methylated flavones (20, 25, 26) and the putative diterpenoids #1–#3 (48–50) showed significant positive correlations between abundance and antifungal activity against most of the fungal species. Subfractions 4-1 to 4-7 showed moderate antifungal activity against individual fungal species. Against *F. culmorum*, subfraction 4-1, 4-2, 4-6 and 4-7 showed the highest antifungal activities, whereas subfraction 4-6 was the most active one against *F. sambucinum*. Mycelial growth of *B. cinerea* was most strongly inhibited by subfractions 4-1, 4-3, and 4-6. The latter fraction also showed the highest antifungal activity against *A. dauci*. Inhibition rates of subfractions 4-1 to 4-7 against *C. lindemuthianum* were generally below 50%, with subfraction 4-2 being the most active.

## 3. Discussion

In the present study, hydroethanolic leaf extracts of 14 populations of *Z. multiflora* were investigated as potential sources of non-volatile semi-polar secondary metabolites with antifungal activity against five important pre- and postharvest fungal pathogens. Due to their sessile lifestyle, land plants in their natural habitats have to cope constantly with a range of abiotic (drought, heat, UV radiation) and biotic (microorganisms, insects) stress factors. The corresponding adaption processes of plants to these stressors often include the formation of specialized metabolites with a broad range of bioactivities [17]. For example, the biosynthesis of flavonoids in plants is upregulated in response to high solar irradiance and excess metal ions [18]. In addition, antimicrobial phytoalexins are synthesized de novo following pathogen attack. As a result of these adaption processes, habitat-specific chemotypes can be formed, which are extremely valuable resources for bioactivity studies and help to fully exploit the metabolic repertoire of a given plant species. The metabolite profiles of unfractionated crude extracts of different chemotypes can be correlated with the different biological activities to identify possible target metabolites for isolation and bioassay. In comparison to the classical, time-consuming, bioassay-guided fractionation techniques, such correlation approaches have the potential to accelerate the process of bioactive metabolite discovery [19].

An in-depth analysis of the semi-polar metabolite profile of leaves of the Konar Siah population revealed the presence of flavones, flavanones, and dihydroflavonols as major flavonoids in *Z. multiflora*. Thus, *Z. multiflora* has a similar spectrum of flavonoids as described for other Nepetoideae plants, such as rosemary, oregano, sage, basil, and thyme [20,21,22]. Apigenin and luteolin, the two main unmethylated flavones, were present mainly as *O*- and *C*-linked mono- and diglycosides. In addition, small amounts of two 6-hydroxyluteolin *O*-glycosides were detected. From a biosynthetic point of view, the co-occurrence of 6-hydroxyapigenin (scutellarein) and 6-hydroxyluteolin would be plausible. However, the scutellarein *O*-glycosides were below the detection limit in leaves of the Konar Siah population but could become detectable when analyzing leaf material of other populations. As in numerous other species of the Nepetoideae, nine different *O*-methylated flavones (17–20, 23–26) with one to three methyl groups were detected, of which four of the mono-*O*-methylated compounds (17–20) could be clearly identified using reference compounds. *O*-Methylation of flavones alters their physico-chemical properties, such as lipophilicity, and protects hydroxy groups from conjugation reactions, such as glycosylation. The identification of genkwanin (17), hispidulin (19-1), chrysoeriol (19-2), and 7-*O*-methylluteolin suggests that *Z. multiflora* has at least the enzymatic ability to methylate flavones in the 7-*O*, 6-*O*, and 3′-*O* positions. Interestingly, *O*-glycosides of chrysoeriol (21) and hispidulin (22) were detected, whereas the two identified 7-*O*-methylated flavones (genkwanin, 7-*O*-methylluteolin) were exclusively detectable in the unconjugated form. This observation supports the hypothesis that *O*-glycosylation of flavones occurs mainly in *Z. multiflora* leaves in the 7-*O* position. Besides flavones, the flavanones naringenin (27), eriodictyol (28), and sakuranetin (29) as well as the dihydroflavonols dihydrokaempferol (31) and dihydroquercetin (32) were identified. Despite a targeted search, no glycoconjugates of flavanones and dihydroflavonols as well as other *O*-methylated derivatives besides sakuranetin were detected in leaf extracts of the Konar Siah population. According to the proposed biosynthetic scheme of (iso)schaftoside [14], 2-hydroxynaringenin 6,8-di-*C*-hexoside (30) could be a biosynthetic precursor of the highly abundant apigenin 6,8-di-*C*-hexoside (4).

Correlation analyses of metabolite abundances with antifungal activities indicated putative antifungal activities of four *O*-methylated flavones (18, 20, 25, 26), two flavanones (27, 28), and two dihydroflavonols (31, 32). In bioassay-guided fractionation, various *O*-methylated flavones were present in subfractions with moderate and high antifungal activity. In particular, the yet uncharacterized di-*O*-methylated and tri-*O*-methylated flavones (25, 26) from subfractions 4-1 and 4-3 showed promising activities against *F. culmorum* and *B. cinerea*. Naringenin (27), eriodictyol (28), and dihydrokaempferol (31) were the major constituents of fraction 3, which was not further fractionated during the bioassay-guided fractionation. Possibly, these three flavonoids could be responsible for the moderate antifungal activity of this fraction. The antifungal activities of some of the identified flavonoids are well-documented in the literature. Naringenin and dihydrokaempferol showed moderate antifungal activity against *Candida albicans* and *Cryptococcus neoformans* [23]. In maize, the biosynthesis of non-*O*-methylated and *O*-methylated flavones, flavanones, and dihydroflavonols is stimulated in the leaves upon infection with *Bipolaris maydis* [24]. The antifungal activity of selected representatives, such as naringenin, genkwanin, and xilonenin against the plant pathogens *Fusarium graminearum*, *Fusarium verticillioides*, and *Rhizopus microspores* has been demonstrated [24]. Similarly, sakuranetin (7-*O*-methylnaringein), detected in subfraction 4-4 with moderate activity against *Fusarium culmorum*, is a major phytoalexin in rice accumulating in blast-infected leaves and was shown to effectively inhibit spore germination of *Pyricularia oryezae* [25].

The annotated HCA derivatives comprise the caffeic acid esters rosmarinic acid (33), chlorogenic acid (39), and nepetoidin (40), as well as a glycosylated coumaric acid derivative (41). As in many other species of the Nepetoideae, rosmarinic acid was the quantitatively dominant HCA derivative in *Z. multiflora* leaf extracts and was accompanied by a number of dehydro-oligomers, including a dehydrodimer (34), two dehydrotrimers (35, 36), and a dehydrotetramer (37). Rosmarinic acid dehydrodimers, such as salvianolic acid B/lithospermic acid B and salvianolic acid E have already been isolated from *Salvia* species [26], but have also been described for other Nepetoideae, such as *Origanum vulgare* [27] and *Melissa officinalis* L. [28]. Higher rosmarinic acid dehydrooligomers have been found, for example, in *Celastrus hindsii* [29]. In order to structurally elucidate the linkage motifs of rosmarinic acid in the detected dehydrooligomers 34–37, their isolation and NMR spectroscopic analysis are required. The detection of nepetoidin in the leaves of *Z. multiflora* supports its role as chemotaxonomic marker to distinguish the Nepetoideae from the other subfamilies of the Lamiaceae [15]. Both nepetoidin A and B showed antifungal activity against *Aspergillus niger* [15]. However, in our study, nepetoidin (40) was not highlighted by correlation analyses or bioassay-guided fractionation. One reason for this could be the chemical instability of the nepetoidins, which accumulate in the glands of the leaf and easily decompose when the plant material dries [15].

Correlation analyses indicated positive associations between the abundance of THQ glycosides 43–47 and antifungal activity. THQ is biosynthesized from geranyl diphosphate, which is cyclized to *γ*-terpinene by a terpene synthase and subsequently oxidized in position 3 or 6 by a cytochrom P450 monooxygenase and a short-chain dehydrogenase/reductase to thymol or carvacrol. The latter are again hydroxylated to THQ by a cytochrom P450 monooxygenase [30]. With regard to the biosynthesis of THQ, it is not surprising that leaves of the populations Siriz and Haneshk, which have a linalool EO chemotype with comparably low levels of carvacrol and thymol [3], also have the lowest levels of THQ (42) and its glycoconjugates 43–47 (Figure 3). Furthermore, rank correlation analyses using metabolite abundances in essential oils and hydroethanolic leaf extracts from all 14 populations revealed positive associations between carvacrol and THQ glycosides 43–47, and between thymol and THQ (42) (Appendix A), supporting their biochemical relationship. Due to their relatively high polarity, THQ glycosides 43–47 were present in fractions 1 and 2 after the first fractionation step, which showed only low antifungal activities in the majority of cases. Despite the positive correlation between abundance and antifungal activity, it is therefore unlikely that THQ glycosides are potent antifungal agents. THQ (42) itself was detected in the most active fraction 4. However, it could not be recovered after the second fractionation step by semi-preparative HPLC and was therefore not tested for its antifungal activity. The reason for this could be the limited chemical stability of THQ. It has been reported that THQ is spontaneously oxidized to thymoquinone when left to stand in hexane at room temperature [30].

Among the eight structurally unidentified diterpenoids, the abundances of diterpenoids #1–#3 (48–50) and diterpenoids #7–#8 (54–55) correlated significantly positively with antifungal activity. Moreover, diterpenoids #1–#3 (48–50) were found in the most active subfractions 4-6 and 4-7. Diterpenoid #1 (48) had the same elemental composition as the phenolic diterpenoid carnosol, but did not co-elute with an authentic carnosol standard. In the constructed debiased sparse partial correlation network, the annotated diterpenoids, with the exception of 51, formed a module in one of the subnetworks (Appendix A), supporting their biosynthetic relationship. Unfortunately, the molecular structures of 48–55 could not be elucidated based on the obtained CID mass spectra. However, the antifungal diterpenoids #1–#3 showed absorption maxima in the range of 273–281 nm and upon CID of the pseudomolecular ions neutral losses of propene, propyl radicals, or propane, which could be associated with aromatic isopropyl groups. Therefore, 48–50 can be assumed to be phenolic diterpenoids with structural similarity to abietane-type phenolic diterpenoids known from other species of the Nepetoideae, such as carnosol and carnosic acid. Interestingly, carnosol and carnosic isolated from *Salvia fruticosa* showed antifungal activity against *Aspergillus tubingensis*, *Botrytis cinerea*, and *Penicillium digitatum* [31]. The antifungal diterpenoids #1–#3 (48–50) are therefore interesting targets for further investigation, including isolation, structure elucidation by NMR, and determination of the spectrum of activity against various plant pathogenic fungi.

The bioassay-guided fractionation revealed other potential antifungal metabolites that were not highlighted by correlation analyses. Although hispidulin (19-1) and chrysoeriol (19-2) showed negative correlations between abundance and antifungal activity, they were present together with apigenin (1) in subfraction 4-2, which showed good antifungal activity against *F. culmorum*. Similarly, genkwanin (17) and sakuranetin (29) were detectable in the moderately active subfractions 4-7 and 4-4, respectively, but showed no significant correlations between abundance and antifungal activity. Previous studies have demonstrated the antifungal activities of hispidulin against *B. cinerea* [31], of chrysoeriol against *Fusarium graminearum* and *Pythium graminicola* [32], and of genkwanin against *Fusarium verticillioides* and *Rhizopus microsporus* [24]. It is therefore likely that these compounds also contribute to the antifungal activity of the hydroethanolic leaf extracts of *Z. multiflora*. In general, bioactive phenolic and flavonoid compounds can prevent fungal growth by inhibiting cell wall formation, cell division, and RNA and protein synthesis [33]. However, the bioactivity of a crude plant extract is a complex trait determined by the specific activities and absolute concentrations of a number of bioactive metabolites. It can additionally be influenced by synergistic effects that cannot be resolved even by bioassay-guided fractionation. Often bioactive metabolites are biosynthetically related to each other, so their accumulation in plant tissue is co-regulated. Based on a diverse set of crude plant extracts with contrasting antifungal activities, correlation analyses can reveal such relationships and accelerate the discovery of antifungal metabolites. However, correlation approaches are always only a first step in identifying bioactive metabolites and cannot replace fractionation approaches and bioassays in providing causative evidence for the bioactivity of a fraction of a crude plant extract or a purified metabolite.

In conclusion, correlation analyses and bioassay-guided fractionation revealed numerous mono-, di-, and tri-*O*-methylated flavones (17, 19, 20, 25, 26), as well as the yet unidentified phenolic diterpenoids 48–50 as promising candidates for metabolites with broad antifungal activity against plant pathogenic fungi. Further studies are now required to determine the molecular structures of the yet unidentified candidate antifungal metabolites and to accurately quantify and compare their antifungal activity and spectrum of activity using pure substances.

## 4. Experimental

### 4.1. Plant Material, Fungal Species and Chemicals

A total of 123 *Z. multiflora* plants were collected at flowering stage in 14 natural habitats across five provinces from the center to the south of Iran, including their major growing areas in the provinces of Isfahan, Kerman, Yazd, Fars, and Hormozgan (Appendix A). Six to eleven individual shrubs were sampled from each habitat. A voucher specimen (No. MPH-1799) was deposited in the Herbarium of Medicinal Plants and Drugs Research Institute, Shahid Beheshti University, Iran. After drying at room temperature (20–25 °C) for 4 days, leaves were harvested. Bioassays were performed with five fungal species. The strains were taken from the mycological culture collection of the BBA/JKI (Berlin, Germany), including *Fusarium culmorum* (JKI-Nr. 62188), *Fusarium sambucinum* (JKI-Nr. 72466), *Botrytis cinerea* (JKI-Nr. 68731), *Alternaria dauci* (JKI-Nr. 70732), and *Colletotrichum lindemuthianum* (JKI-Nr. 71735). Ethanol (≥99.9%, for HPLC), methanol (≥99.95% for LC-MS) and acetonitrile (≥99.95% for LC-MS) were supplied by Th. Geyer (CHEMSOLUTE). Formic acid (≥98%, for LC-MS) was purchased from Sigma-Aldrich. Ultrapure water (resistivity ≥ 18.2 MΩ cm) was obtained from an Arium 611 water purification system (Sartorius). Sources of reference compounds used for metabolite identification are listed in Appendix A.

### 4.2. Preparation of Hydroethanolic Leaf Extracts for Bioassays and Metabolite Analyses

Dried leaf material (approx. 3–4 g) from each of the 123 samples was ground to a fine powder (5 min at 30 s^−1^) using a mixer mill (Retsch MM2) and a steel ball (∅ 8 mm). To reduce the number of samples for bioassays and metabolite analyses, equal aliquots of homogenized leaf material of 2–4 individual shrubs from the same habitat were combined and thoroughly mixed, resulting in a total of 42 pooled leaf samples (14 habitats × 3 pooled samples). Homogenized leaf material (2.50 g) was weighed into a 15-mL polypropylene centrifuge tube. After addition of 10 mL ethanol/water, 1/1 (*v*/*v*), the mixture was vortex-mixed (1 min), sonicated (15 min, 20–25 °C), and shaken (30 min, 2000 min^−1^, room temperature). The supernatant was transferred into a 25-mL volumetric flask after centrifugation (10 min, 4696× *g*, 20 °C). The remaining residue was extracted once again with 10 mL ethanol/water, 1/1 (*v*/*v*), as described above. Both supernatants were combined, and their volume was adjusted to 25 mL using ethanol/water, 1/1 (*v*/*v*). The resulting stock extract (100 mg dry leaf material per mL extract) was stored at 6 °C in the fridge until further use in antifungal assays on 9 cm Petri dishes. For LC/MS-based metabolite profiling, an aliquot of the stock extract was diluted 1/50 with ethanol/water, 1/1 (*v*/*v*). For preparation of a quality control (QC) sample, equal aliquots of each of the 42 diluted stock extracts were pooled.

### 4.3. Antifungal Assays

Antifungal assays were performed according to the agar well diffusion method with slight modifications [34,35]. Potato dextrose agar (approx. 20 mL) was poured into Petri dishes (∅ 9 cm). After solidification, an agar plug (∅ 6 mm) was removed using a sterile cork borer. The resulting hole was filled with 100 µL hydroethanolic leaf extract (see Section 4.2), 100 µL dissolved extract fraction (see Section 4.4), or 100 µL ethanol/water, 1/1 (*v*/*v*) as control. Afterwards, an agar plug (∅ 6 mm) containing actively growing mycelia was excised from a fungal preculture and placed onto the agar at a distance of 4 cm from the punched hole. Petri dishes were incubated for 4–7 days at 20 °C in darkness and radial growth was monitored every 24 h. Antifungal activity was evaluated by measuring the growth of the mycelia in the direction of the punched hole [34]. The inhibition rate (IR) was calculated using the formula, IR (%) = [(C − T)/C] × 100, where C is the average mycelial growth of the control and T the mycelial growth of the treatment. All assays were performed in triplicate.

In case of limited substance amounts, antifungal assays were performed in smaller Petri dishes (∅ 4 cm) filled with 8 mL of potato dextrose agar. The diameter of the agar hole and the inoculum was reduced to 4 mm, and the assay was run with 30 µL test solution.

### 4.4. Fractionation of Hydroethanolic Leaf Extracts

Equal aliquots of air-dried and homogenized leaf material from all samples collected in the Konar Siah habitat were pooled. The resulting sample (10 g) was extracted twice with 100 mL ethanol/water, 1/1 (*v*/*v*), following the method described in Section 4.2. Both extracts were combined and evaporated to dryness *in vacuo* to give 2.95 g crude extract. An aliquot of the crude extract (1.50 g) was solubilized in 150 mL of methanol/water, 2/8 (*v*/*v*) and subjected to solid-phase extraction (SPE) using six Strata C18-E Giga Tubes (10 g/60 mL, Phenomenex, Torrance, CA, USA). SPE cartridges were conditioned with 60 mL of methanol and equilibrated with 60 mL of methanol/water, 2/8 (*v*/*v*). Afterwards, 25 mL of the solubilized leaf extract and 25 mL of methanol/water, 2/8 (*v*/*v*), were successively applied on the cartridge and the eluate was collected (fraction 1). The cartridge was then eluted with 50 mL methanol/water, 4/6 (*v*/*v*), 50 mL methanol/water, 6/4 (*v*/*v*), 50 mL methanol/water, 8/2 (*v*/*v*), and 50 mL methanol to give fractions 2 to 5, respectively. Fractions of individual SPE cartridges were combined and evaporated to dryness *in vacuo* yielding 868 mg of fraction 1, 402 mg of fraction 2, 114 mg of fraction 3, 48 mg of fraction 4, and 60 mg of fraction 5. Aliquots of the crude extract and of fractions 1 to 5 were dissolved in ethanol/water, 1/1 (*v*/*v*) at a concentration of 10 mg/mL and subjected to antifungal assays on 9 cm Petri dishes.

For further fractionation, 30 mg of fraction 4 was dissolved in 1.3 mL methanol/water, 8/2 (*v*/*v*) and subjected to semi-preparative HPLC, which was performed on an 1100 Series HPLC system (Agilent Technologies, St. Clara, CA, USA) equipped with a ReproSil XR 120 C18 column (250 × 8 mm, 4 μm particle size, Dr. Maisch, Ammerbuch, Germany). Water and methanol were both acidified with 0.1% (*v*/*v*) formic acid and used as eluents A and B, respectively. The following binary gradient program at a flow rate of 3 mL min^−1^ was applied: 0–1 min, isocratic 1% B; 1–1.5 min, linear from 1% to 35% B; 1.5–17 min, linear from 35% to 75% B, 17–17.5 min, linear from 75% to 100% B; 17.5–20 min, isocratic 100% B. The column temperature was maintained at 40 °C. Eluting compounds were monitored with a diode array detector at 280 nm and 340 nm. A total of 24 separations were carried out using an injection volume of 50 μL. Seven fractions were collected in the retention time range between 12.5 and 18.5 min. After removal of volatiles compounds *in vacuo*, 0.2 mg of fraction 4-1, 1.3 mg of fraction 4-2, 0.6 mg of fraction 4-3, 0.9 mg of fraction 4-4, 1.4 mg of fraction 4-5, 2.4 mg of fraction 4-6, and 1.8 mg of fraction 4-7 were obtained. Fractions 4-1 to 4-7 were dissolved in ethanol/water, 1/1 (*v*/*v*) at a concentration of 1 mg/mL and subjected to antifungal assays on 4 cm Petri dishes and after further dilution to LC/MS analysis.

### 4.5. UHPLC/DAD/ESI-QTOFMS

LC/MS analyses were performed on an Infinity 1290 series UHPLC system (Agilent Technologies) consisting of a binary pump (G4220A), an autosampler (G4226A, 20 µL loop), an autosampler thermostat (G1330B), and a thermostatted column compartment (G1316C) which was coupled in series with a diode array detector (G4212B) and an iFunnel Q-TOF mass spectrometer (G6550A, Agilent Technologies) via a dual Agilent jet stream electrospray ion source. MassHunter LC/MS Data Acquisition software (Agilent Technologies, version B.06.01) was used for controlling the instrument and data acquisition as well as MassHunter Qualitative and Quantitative Analysis software (Agilent Technologies, version B.07.00) for data evaluation. The mass spectrometer was operated in a low mass range (*m*/*z* 1700) and an extended dynamic range (2 GHz) mode. Using these settings, the mass resolution (full width at half maximum) at *m*/*z* 922 was approx. 23,000. The instrument was autotuned and calibrated according to manufacturer´s recommendations using ESI-L tuning mix (Agilent Technologies). Reference mass correction was used throughout all experiments. For this purpose, a solution of purine (20 µM) and hexakis-(2,2,3,3-tetrafluoropropoxy)phosphazine (20 µM) in acetonitrile/water, 95/5 (*v*/*v*) was continuously introduced through the second sprayer of the dual ion source at a flow rate of 20 µL min^−1^ using an external HPLC pump equipped with a 1:100 splitting device.

Extracts (1 µL) were separated on a Zorbax RRHD Eclipse Plus C18 column (100 mm × 2.1 mm, 1.8 µm particle size, Agilent Technologies) using water and methanol both acidified with 0.1% (*v*/*v*) formic acid as eluent A and B, respectively. The following binary gradient program at a flow rate of 400 µL min^−1^ was applied: 0–10 min, linear from 5% to 95% B; 10–13 min, isocratic, 95% B; 13–15 min, isocratic, 5% B. The column temperature was maintained at 40 °C and the autosampler temperature was kept at 6 °C. Eluting compounds were sequentially detected in a wavelength range of 190–600 nm and in an *m*/*z* range of 70–1700, either in positive or negative ion mode. Absorption spectra were acquired using an acquisition rate of 2.5 spectra per second, and centroid mass spectra using an acquisition rate of 3 spectra per second. Instrument settings were as described by Tais et al. [36].

To correct for systematic instrumental drift within metabolite profiling experiments, pooled QC samples were repeatedly analyzed after six analytical samples as well as in the beginning and the end of the experiment. CID mass spectra were acquired in targeted-MS^2^ mode using scheduled precursor ion lists and the following parameters: acquisition rate MS, 3 spectra per second; acquisition rate MS/MS, 3 spectra per second; isolation width, narrow (1.3 *m*/*z*); collision energy, 10, 20, 30, and 40 V; collision gas, nitrogen. For acquisition of CID mass spectra of in-source fragment ions (pseudo-MS^3^), funnel exit DC voltage was increased from 50 to 120 V.

### 4.6. LC/MS Data Processing

Raw data files were converted into *mzData* format using MassHunter Qualitative Analysis software, arranged separately for each ion mode in 14 sample classes, and processed using the R package “XCMS” (version 3.11.3) [37]. Feature detection was carried out using the centWave algorithm [parameters: prefilter = (3, 1000); sntresh = 3; ppm = 25; peak width = (5, 12)]. Alignment was accomplished by consecutive application of the functions *group.density* (parameters: minfrac = 1; bw = 2; mzwid = 0.02), *retcor.loess* (parameters: span = 1; missing = 0; extra = 0) and *group.density* (parameters: minfrac = 1; bw = 1.5, mzwid = 0.02). Missing feature intensities were replaced by evenly distributed random numbers in the range of 500–600 as an intensity threshold. The resulting feature intensity matrices (12,783 features × 42 samples for positive ion mode, 10,517 features × 42 samples for negative ion mode) were subjected to unsupervised random forest classification and to ANOVA.

For relative quantification of annotated metabolites 1–67, extracted ion chromatograms were generated for respective quantifier ions (Appendix A, *m*/*z* width 0.02) and integrated using MassHunter Quantitative Analysis software. Systematic instrumental drift was corrected individually for each metabolite using repeatedly injected pooled QC samples and a LOWESS/Spline interpolation algorithm (http://prime.psc.riken.jp/compms/others/main.html#Lowess, accessed on 26 April 2022). The obtained signal-drift corrected peak areas were used for ANOVA, for construction of a heatmap and a debiased sparse partial correlation network, as well as for rank correlation with antifungal activities.

### 4.7. Statistical Analysis

Statistical analyses were conducted using R statistical software (R Foundation for Statistical Computing, Vienna, Austria). To control the false discovery rate in multiple testing, *p* values were adjusted using the Benjamini–Hochberg procedure implemented in the function *p.adjust* from the package “stats” (version 4.1.0). The function *anova* from the same package was used for variance analyses. Metabolite abundances were log_2_-transformed prior to ANOVA. Fisher´s least significance difference test was performed with the function *LSD.test* from the package “agricolae” (version 1.3-5). Unsupervised random forest classification was performed using the function *randomForest* from the package “randomForest” (version 4.7-1.1). The obtained proximity matrix was visualized by multidimensional scaling using the function *cmdscale* from the package “stats”. Spearman´s rank correlation coefficients and corresponding *p* values were calculated using the function *rcorr* from the package “Hmisc” (version 4.7-0). The heatmap was generated in MS Excel 2016 (Microsoft, Redmond, WA, USA) using conditional formatting. For this purpose, signal-drift corrected peak intensities of quantifier ions of metabolites 1–67 were averaged for each population, normalized to the mean of all populations and log_2_-transformed. A debiased sparse partial correlation network was generated from log_2_-transformed instrumental-drift-corrected peak areas of quantifier ions of metabolites 1–67 using MetaboAnalyst (https://www.metaboanalyst.ca, accessed on 21 June 2022).

## Figures and Tables

**Figure 1 molecules-27-08903-f001:**
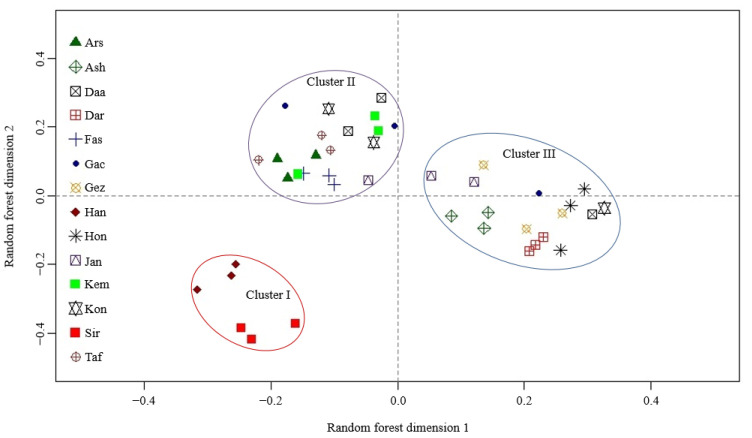
Unsupervised random forest classification of metabolite profiles obtained from hydroethanolic *Z. multiflora* leaf extracts (14 populations × 3 samples) by reversed-phase UHPLC/DAD/ESI-QTOFMS operated in positive ion mode. The scatter plot obtained by multidimensional scaling of the proximity matrix is shown.

**Figure 2 molecules-27-08903-f002:**
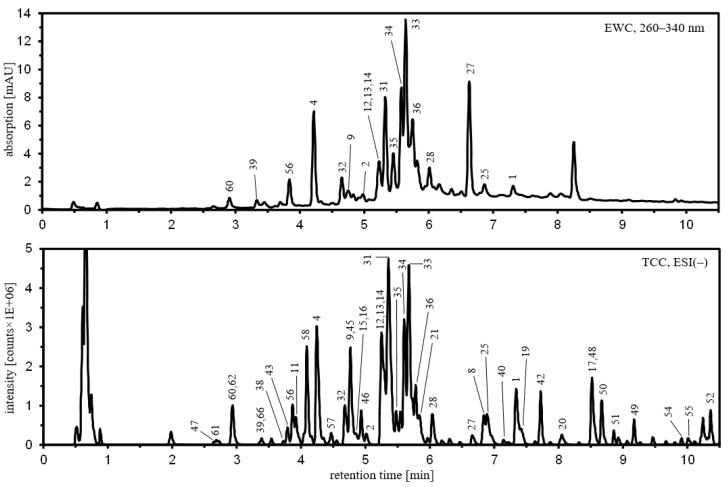
Representative chromatograms obtained from a hydroethanolic extract of a pooled leaf sample of the Konar Siah population using UHPLC/DAD/ESI-QTOFMS operated in negative ion mode. The upper panel shows an extracted wavelength chromatogram (EWC) from 260–340 nm, the lower panel a total compound chromatogram (TCC), which was reconstructed from raw data using the algorithm “Find Compounds by Molecular Feature” implemented in MassHunter Qualitative Data Analysis. For compound labelling, see Table 2. Further analytical data is detailed in Appendix A.

**Figure 3 molecules-27-08903-f003:**
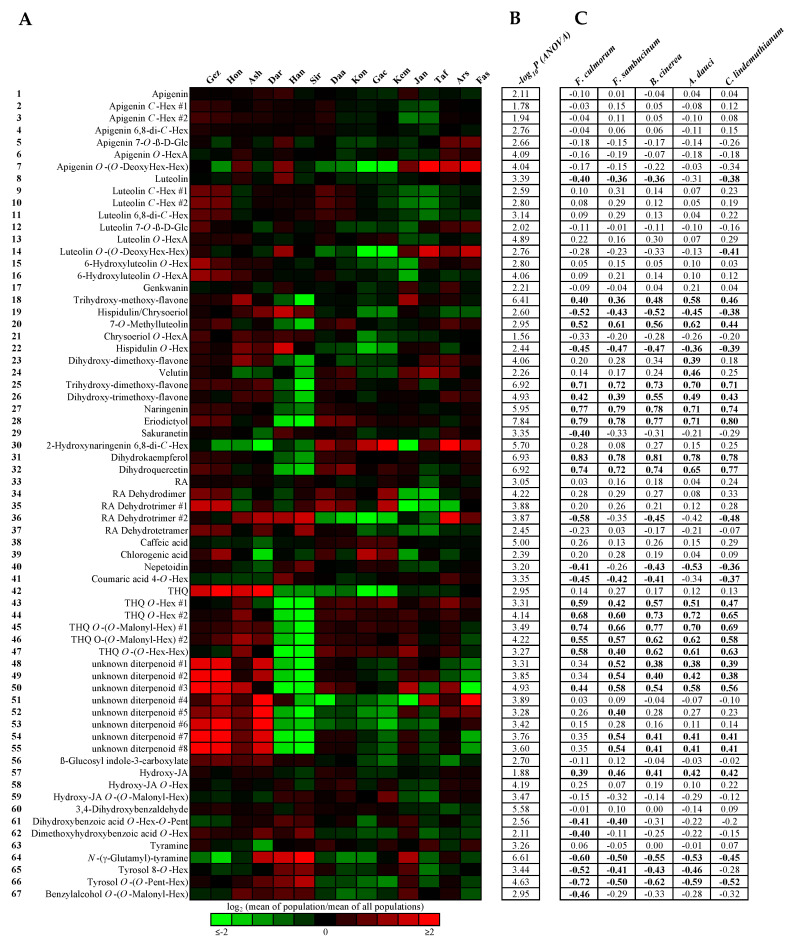
Heatmap representation of mean abundances of metabolites 1-67 in leaves of 14 *Z. multiflora* populations (**A**). Adjusted *p* values from variance analyses of metabolite abundances with population as factor (**B**). Spearman’s rank correlation coefficients calculated from metabolite abundances and fungal inhibition rates (*n* = 42). Statistically significant rank correlations (*p*_adj_ ≤ 0.05) are marked in bold (**C**).

**Figure 4 molecules-27-08903-f004:**
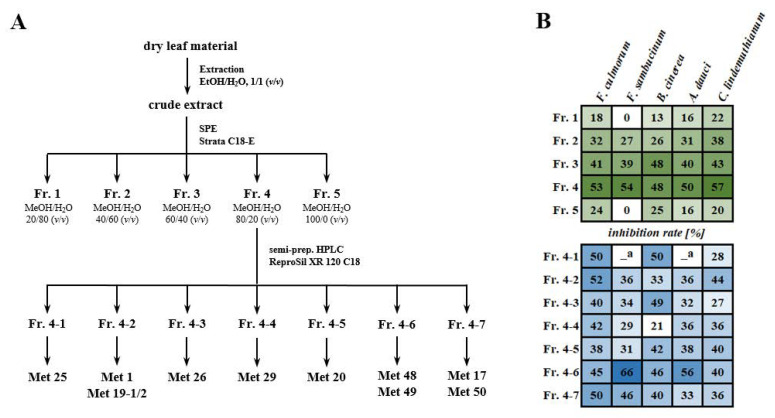
Schematic representation of the bioassay-guided fractionation of a hydroethanolic extract of a pooled leaf sample of the Konar Siah population (**A**); antifungal effects of the fractions obtained (**B**). The antifungal assays of fractions 1 to 5 were performed in 9 cm Petri dishes using test solutions in ethanol/water, 1/1 (*v*/*v*) with a concentration of 10 mg/mL. Due to limited substance amounts, fractions 4-1 to 4-7 were assayed in 4 cm Petri dishes at a concentration of 1 mg/mL. Shown inhibition rates are means of three replicates and reflected by a heatmap. Darker green or blue colors correspond to higher inhibition rates. The complete data set and statistical analyses are given in Appendix A. ^a^ Assay not performed due to limited substance amounts.

**Table 1 molecules-27-08903-t001:** Antifungal activities of hydroethanolic leaf extracts from plants of 14 *Z. multiflora* populations against five phytopathogenic fungi. Mean growth inhibition rates and standard deviations determined from three leaf samples per population are shown. Mean values in a column that do not have a common letter are significantly different (Fisher’s least significant difference test, *p*_adj_ ≤ 0.05).

Population	Code	Inhibition of Mycelial Growth (%)				
		*F. culmorum*	*F. sambucinum*	*B. cinerea*	*A. dauci*	*C. lindemuthianum*
Arsenjan	Ars	51.5 ± 2.5 ^a,b,c^	50.3 ± 3.8 ^a,b,c^	50.5 ± 1.5 ^a,b,c^	58.7 ± 3.1 ^b^	48.2 ± 5.5 ^d,e^
Ashkezar	Ash	49.3 ± 3.1 ^b,c^	40.5 ± 3.3 ^d^	46.1 ± 1.4 ^b,c,d^	55.1 ± 1.5 ^b,c^	44.8 ± 2.8 ^e^
Daarbast	Daa	57.5 ± 2.5 ^a^	51.4 ± 3.6 ^a,b^	53.9 ± 3.7 ^a^	58.1 ± 3.9 ^b^	65.7 ± 4.2 ^a^
Darab	Dar	52.2 ± 2.9 ^a,b,c^	46.4 ± 2.3 ^b,c,d^	44.2 ± 2.3 ^c,d,e^	51.6 ± 1.1 ^c,d^	47.6 ± 2.7 ^e^
Fasa	Fas	52.9 ± 2.8 ^a,b,c^	43.2 ± 1.5 ^b,c,d^	40.1 ± 7.7 ^d,e^	48.4 ± 1.7 ^d^	36.1 ± 3.7 ^f^
Gachooyeh	Gac	54.6 ± 0.7 ^a,b,c^	47.6 ± 7.4 ^b,c,d^	46.9 ± 1.6 ^a,b,c,d^	53.5 ± 1.3 ^b,c,d^	46.2 ± 2.2 ^e^
Gezeh	Gez	56.2 ± 3.4 ^a,b^	48.4 ± 2.7 ^a,b,c,d^	50.9 ± 1.5 ^a,b,c^	58.3 ± 4.7 ^b^	55.4 ± 2.7 ^c^
Haneshk	Han	36.8 ± 2.9 ^d^	24.4 ± 10.5 ^e^	21.7 ± 5.4 ^f^	36.7 ± 2.1 ^e^	22.5 ± 2.3 ^g^
Hongooyeh	Hon	54.5 ± 3.3 ^a,b,c^	57.6 ± 2.6 ^a^	51.2 ± 2.3 ^a,b,c^	56.4 ± 3.5 ^b,c^	48.7 ± 3.7 ^d,e^
Jandaq	Jan	57.3 ± 3.4 ^a^	50.9 ± 2.3 ^a,b,c^	54.4 ± 0.8 ^a^	64.9 ± 3.5 ^a^	62.6 ± 3.3 ^a,b^
Kemeshk	Kem	56.6 ± 8.9 ^a,b^	43.3 ± 6.4 ^b,c,d^	50.9 ± 4.2 ^a,b,c^	53.6 ± 3.4 ^b,c,d^	53.8 ± 4.1 ^c,d^
Konar Siah	Kon	59.1 ± 1.9 ^a^	51.7 ± 3.7 ^a,b^	51.9 ± 1.7 ^a,b^	59.6 ± 3.3 ^a,b^	59.1 ± 0.6 ^b,c^
Siriz	Sir	31.4 ± 3.6 ^d^	29.6 ± 3.8 ^e^	18.9 ± 8.3 ^f^	32.7 ± 4.8 ^e^	21.1 ± 3.6 ^g^
Taft	Taf	47.9 ± 3.8 ^c^	41.3 ± 4.5 ^c,d^	37.9 ± 0.5 ^e^	54.1 ± 3.4 ^b–d^	36.1 ± 4.2 ^f^

**Table 2 molecules-27-08903-t002:** Annotated metabolites detected in hydroethanolic leaf extracts of *Z. multiflora*, Konar Siah population. For tandem mass spectral data and further information on annotation, see Appendix A.

No.	Compound	Elemental Composition	AL ^a^	RT [min]	Quantifier Ion		
					Type	*m/z*	*m/z*
						Measured	Calculated
1	Apigenin	C_15_H_10_O_5_	1	7.34	[M + H]^+^	271.0606	271.0601
2	Apigenin *C*-Hex—isomer#1	C_21_H_20_O_10_	2	5.01	[M + H]^+^	433.1128	433.1129
3	Apigenin *C*-Hex—isomer#2	C_21_H_20_O_10_	2	5.25	[M + H]^+^	433.1131	433.1129
4	Apigenin 6-*C*-Hex-8-*C*-Hex	C_27_H_30_O_15_	2	4.23	[M + H]^+^	595.1656	595.1657
5	Apigenin 7-*O*-*β*-d-Glc	C_21_H_20_O_10_	1	5.75	[M + H]^+^	433.1125	433.1129
6	Apigenin *O*-HexA	C_21_H_18_O_11_	2	5.72	[M + H]^+^	447.0923	447.0922
7	Apigenin *O*-(*O*-DeoxyHex-Hex)	C_27_H_30_O_14_	2	5.65	[M + H]^+^	579.1699	579.1708
8	Luteolin	C_15_H_10_O_6_	1	6.82	[M + H]^+^	287.0552	287.0550
9	Luteolin *C*-Hex—isomer#1	C_21_H_20_O_11_	2	4.69	[M + H]^+^	449.1073	449.1078
10	Luteolin *C*-Hex—isomer#2	C_21_H_20_O_11_	2	4.76	[M + H]^+^	449.1075	449.1078
11	Luteolin 6-*C*-Hex-8-*C*-Hex	C_27_H_30_O_16_	2	3.92	[M + H]^+^	611.1606	611.1607
12	Luteolin 7-*O*-*β*-d-Glc	C_21_H_20_O_11_	1	5.27	[M + H]^+^	449.1079	449.1078
13	Luteolin *O*-HexA	C_21_H_18_O_12_	2	5.23	[M + H]^+^	463.0871	463.0871
14	Luteolin *O*-(*O*-DeoxyHex-Hex)	C_27_H_30_O_15_	2	5.25	[M + H]^+^	595.1657	595.1657
15	6-Hydroxyluteolin *O*-Hex	C_21_H_20_O_12_	2	4.84	[M − H]^−^	463.088	463.0882
16	6-Hydroxyluteolin *O*-HexA	C_21_H_18_O_13_	2	4.8	[M + H]^+^	479.0818	479.082
17	Genkwanin	C_16_H_12_O_5_	1	8.55	[M − H]^−^	283.0612	283.0612
18	Trihydroxy-methoxy-flavone	C_16_H_12_O_6_	3	6.8	[M − H]^−^	299.0561	299.0561
19-1 ^b^	Hispidulin	C_16_H_12_O_6_	1	7.41	[M − H]^−^	299.0561	299.0561
19-2 ^b^	Chrysoeriol						
20	7-*O*-Methylluteolin	C_16_H_12_O_6_	1	8.06	[M − H]^−^	299.0561	299.0561
21	Chrysoeriol *O*-HexA	C_22_H_20_O_12_	2	5.83	[M − H]^−^	475.0883	475.0882
22	Hispidulin *O*-Hex	C_22_H_22_O_11_	2	5.97	[M − H]^−^	461.1087	461.1089
23	Dihydroxy-dimethoxy-flavone	C_17_H_14_O_6_	3	7.99	[M + H]^+^	315.0862	315.0863
24	Velutin	C_17_H_14_O_6_	2	8.54	[M − H]^−^	313.0712	313.0718
25	Trihydroxy-dimethoxy-flavone	C_17_H_14_O_7_	3	6.87	[M + H]^+^	331.0815	331.0812
26	Dihydroxy-trimethoxy-flavone	C_18_H_16_O_7_	3	7.46	[M + H]^+^	345.0969	345.0969
27	Naringenin	C_15_H_12_O_5_	1	6.66	[M + H]^+^	273.0762	273.0757
28	Eriodictyol	C_15_H_12_O_6_	1	6.03	[M + H]^+^	289.0707	289.0707
29	Sakuranetin	C_16_H_14_O_5_	2	7.93	[M + H]^+^	287.0913	287.0914
30	2-Hydroxynaringenin 6-*C*-Hex-8-*C*-Hex	C_27_H_32_O_16_	2	3.47	[M − H]^−^	611.1604	611.1618
31	Dihydrokaempferol	C_15_H_12_O_6_	1	5.34	[M + H]^+^	289.0709	289.0707
32	Dihydroquercetin	C_15_H_12_O_7_	1	4.66	[M − H]^−^	303.0511	303.051
33	Rosmarinic Acid (RA)	C_18_H_16_O_8_	1	5.69	[M − H]^−^	359.0773	359.0772
34	RA Dehydrodimer	C_36_H_30_O_16_	3	5.6	[M − H]^−^	717.1461	717.1461
35	RA Dehydrotrimer—isomer#1	C_54_H_44_O_24_	3	5.48	[M − H]^−^	1075.2156	1075.215
36	RA Dehydrotrimer—isomer#2	C_54_H_44_O_24_	3	5.78	[M − H]^−^	1075.2149	1075.215
37	RA Dehydrotetramer	C_72_H_58_O_32_	3	5.85	[M − 2H]^2−^	716.1388	716.1383
38	Caffeic acid	C_9_H_8_O_4_	1	3.73	[M − H]^−^	179.0349	179.0350
39	Chlorogenic acid	C_16_H_18_O_9_	1	3.36	[M − H]^−^	353.0873	353.0878
40	Nepetoidin	C_17_H_14_O_6_	2	7.15	[M − H]^−^	313.0715	313.0718
41	Coumaric acid 4-*O*-Hex	C_15_H_18_O_8_	2	3.47	[M − H]^−^	325.0916	325.0929
42	Thymohydroquinone (THQ)	C_10_H_14_O_2_	2	7.73	[M − H]^−^	165.0923	165.0921
43	THQ *O*-Hex—isomer#1 (Zataroside)	C_16_H_24_O_7_	2	3.79	[M + HCOO]^−^	373.1504	373.1504
44	THQ *O*-Hex—isomer#2 (Zataroside)	C_16_H_24_O_7_	2	4.04	[M + HCOO]^−^	373.1499	373.1504
45	THQ *O*-(*O*-Malonyl-Hex)—isomer#1	C_19_H_26_O_10_	2	4.77	[M − H − CO_2_]^−^	369.1553	369.1555
46	THQ *O*-(*O*-Malonyl-Hex)—isomer#2	C_19_H_26_O_10_	2	4.93	[M − H − CO_2_]^−^	369.1552	369.1555
47	THQ *O*-(*O*-Hex-Hex)	C_22_H_34_O_12_	2	2.65	[M + HCOO]^−^	535.2027	535.2032
48	unknown diterpenoid#1	C_20_H_26_O_4_	4	8.52	[M − H]^−^	329.1756	329.1758
49	unknown diterpenoid#2	C_20_H_24_O_4_	4	9.18	[M − H]^−^	327.1600	327.1602
50	unknown diterpenoid#3	C_20_H_26_O_3_	4	8.68	[M − H]^−^	313.1808	313.1809
51	unknown diterpenoid#4	C_20_H_26_O_3_	4	8.86	[M − H]^−^	313.1808	313.1809
52	unknown diterpenoid#5	C_20_H_26_O_3_	4	10.37	[M − H]^−^	313.1808	313.1809
53	unknown diterpenoid#6	C_20_H_24_O_3_	4	10.63	[M − H]^−^	311.1653	311.1653
54	unknown diterpenoid#7	C_20_H_24_O_5_	4	9.91	[M − H]^−^	343.1548	343.1551
55	unknown diterpenoid#8	C_20_H_24_O_5_	4	10.02	[M − H]^−^	343.1548	343.1551
56	*β*-Glucosyl indole-3-carboxylate	C_15_H_17_NO_7_	2	3.87	[M − H]^−^	322.0932	322.0932
57	Hydroxyjasmonic acid (Hydroxy-JA)	C_12_H_18_O_4_	3	4.45	[M − H]^−^	225.1132	225.1132
58	Hydroxy-JA *O*-Hex	C_18_H_28_O_9_	3	4.08	[M − H]^−^	387.1659	387.1661
59	Hydroxy-JA *O*-(*O*-Malonyl-Hex)	C_21_H_30_O_12_	3	4.64	[M − H − CO_2_]^−^	429.1759	429.1766
60	3,4-Dihydroxybenzaldehyde	C_7_H_6_O_3_	1	2.94	[M − H]^−^	137.0247	137.0244
61	Dihydroxybenzoic acid *O*-Hex-*O*-Pent	C_18_H_24_O_13_	3	2.73	[M − H]^−^	447.1141	447.1144
62	Dimethoxyhydroxybenzoic acid *O*-Hex	C_15_H_20_O_10_	3	2.94	[M − H]^−^	359.0981	359.0984
63	Tyramine	C_8_H_11_NO	1	0.88	[M + H − NH_3_]^+^	121.0643	121.0648
64	*N*-(*γ*-Glutamyl)-tyramine	C_13_H_18_N_2_O_4_	2	2.43	[M + H]^+^	267.1337	267.1339
65	Tyrosol 8-*O*-Hex	C_14_H_20_O_7_	2	2.98	[M + NH_4_]^+^	318.1543	318.1547
66	Tyrosol *O*-(*O*-Pent-Hex)	C_19_H_28_O_11_	2	3.39	[M − H]^−^	431.1556	431.1559
67	Benzylalcohol *O*-(*O*-Malonyl-Hex)	C_16_H_20_O_9_	2	4.74	[M + NH_4_]^+^	374.1440	374.1446

^a^ Annotation level: 1: metabolite identified using a reference compound, 2: metabolite putatively annotated based on interpretation of mass spectral data, 3: metabolite class putatively annotated based on interpretation of mass spectral data, 4: unknown metabolite. ^b^ Hispidulin and chrysoeriol coelute under chromatographic conditions used for metabolite profiling, for chromatographic separation of both metabolites a modified eluent system is required (see Appendix A).

## Data Availability

Not applicable.

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
