# Peer review of "Metabolite Profiling and Bioassay-Guided Fractionation of Zataria multiflora Boiss. Hydroethanolic Leaf Extracts for Identification of Broad-Spectrum Pre and Postharvest Antifungal Agents"

_molecules, 2022, doi:10.3390/molecules27248903_

Round 1

Reviewer 1 Report

The experimental work and manuscript preparation of the study was good for the journal. However, I have one question: in order to display the results in the clearer manner, the flow chart of the study should be added into the manuscript.

Author Response

A: Thank you very much for your suggestion. We had provided a graphical abstract that shows the flow chart of the manuscript. In addition, Figure 4 represents the flow chart of the study.

Reviewer 2 Report

The research content is some routine work with relatively low innovation. This manuscript does not fulfill the standards established for the journal to be considered for publication.

Author Response

A: Sorry, we see this different. The volatile compounds of Z. multiflora are known and have been reported before. However, in the present manuscript we report, for the first time, 68 non-volatile compound of Z. multiflora hydroethanolic extracts and the antifungal activity of these extracts. The results provide a valuable information for this species and future studies on non-volatile antifungal compounds from medical plants.

Reviewer 3 Report

1 The use of punctuation needs to be improved. For instance, a comma should be inserted between "regulation" and "the" in line 60;

2 Reference(s) should be provided in line 42;

3 The conclusion section is lengthy and wordy.

Author Response

Thank you for your suggestions.

1) The use of punctuation needs to be improved. For instance, a comma should be inserted between "regulation" and "the" in line 60;

A: The punctuation all over the manuscript had been taken over by an expert.

2) Reference(s) should be provided in line 42;

A: Information provided.

3) The conclusion section is lengthy and wordy.

A: We identified 68 non-volatile compound of Z. multiflora and up to now, there is no publication that reported all these compounds in Z. multiflora extract. Therefore we decided to provide the necessary information to the reader being aware that the discussion is a little longer than usual.

Reviewer 4 Report

Dear Editor, 

I enjoyed reading the article entitled "Metabolite profiling and bioassay-guided fractionation of Zataria multiflora Boiss. hydroethanolic leaf extracts for identification of broad-spectrum pre- and postharvest antifungal agents" submitted for possible publication in Molecules. The manuscript contains interesting information about the phytochemical contents and antifungal activities of Zataria multiflora Boiss. collected from different regions in the south of Iran.  I would suggest the following (minor) alterations/improvements be considered:

Authors are invited to add the main results of the antifungal activity of the most active extract 

Author Response

A: Thank you very much for your comments. The results of the antifungal activity of the active extract are presented in Table 1. In addition, the results of most active fractions are presented in Figure 4 and also in Table S2 in more detail.

Reviewer 5 Report

Reviewer comments:

The manuscript “Metabolite profiling and bioassay-guided fractionation of Za-taria multifloraBoiss. hydroethanolic leaf extracts for identification of broad-spectrum pre- and postharvest antifungal agents” is not acceptable in its present format. The decision over the manuscript is “Major Revision”. All the required corrections are highlighted inside the manuscript with attached comment boxes.

Comments:

1.      Title:

The symbol "-" should be removed from the title.

2.      Keywords:

Arrange all the keywords in alphabetical order.

3.      Introduction:

Page 1, Line number 27, 28: Need reference for this statement.

4.      Introduction:

Page 2, Line number 67-78: The gap of the study was not well established. There is no novelty found in this research. Why this study is needed?? Additionally, there was not much detail was provided about the plant species traditional knowledge in the introduction section.

5.      Results: Page 2, Line number 81, 82:

What’s the different between geographically and environmentally diverse??

6.      Table 2:

Serial no 43,44,45: Author could have written it as Zataroside A and Zataroside B as well.

7.      Table 2:

Serial no 48-55: What about all this diterpenoids?? Will they be identified or will be left??

8.      Page 7, Line number 178:

Where is the reference of authentic reference compounds??

9.      Discussion: Page 12, Line number 380-385:

Simplify the statement. It seems to be little confusing.

10.  Page 12, Line number 389-390:

These are classes of compounds. Did the author found similarity among the species in their respective compounds also. Need clarifications in this aspect. If they have found similar compounds then mention the names.

11.  Page 12, Line number 425-426:

Where are the references for well documented antifungal activity of flavonoids?? Need to cite reference and need to discuss.

12.  Page 13, Line number 428-435:

Probable mechanism of action needs to be provided herewith the discussion section for antifungal activity.

13.  Experimental: Page 14, Line number 527-528:

Where is the latitude, longitude, altitude of the collection sites??

14.  Page 14, Line number 529-531:

Who exactly did the identification of the plant species used for the research??

15.  Page 15, Line number 567-568:

Why did not the author used incubator??

16.  Page 15, Line number 569-572:

Write the formula in their desired form.

The used formula and for the assay as well author need to cite one more valid reference.

17.  Additionally, the language of the manuscript needs to be improved. There are some grammatical and typical mistakes are there. The manuscript needs to be revised by a language expert.

Author Response

The manuscript “Metabolite profiling and bioassay-guided fractionation of Za-taria multifloraBoiss. hydroethanolic leaf extracts for identification of broad-spectrum pre- and postharvest antifungal agents” is not acceptable in its present format. The decision over the manuscript is “Major Revision”. All the required corrections are highlighted inside the manuscript with attached comment boxes.

A: Thank you very much for the helpful suggestions. We took all comments into account.

  1. Title: The symbol "-" should be removed from the title. A: Changed.
  2. Keywords: Arrange all the keywords in alphabetical order. A: Changed.
  3. Introduction: Page 1, Line number 27, 28: Need reference for this statement. A: Information provided.
  4. Introduction: Page 2, Line number 67-78: The gap of the study was not well established. There is no novelty found in this research. Why this study is needed?? Additionally, there was not much detail was provided about the plant species traditional knowledge in the introduction section.

A: We mentioned, the non-volatile secondary metabolites from Z. multiflora are hardly explored (lines 46-47) and that the majority of publications are based on essential oil compounds and their antimicrobial activity. Up to now, there is no study that provides comprehensive semi-polar metabolite profiling of Z. multiflora. Only few studies identified a limited number of semi-polar metabolite of Z. multiflora, whereas we identified 68 non-volatile compounds.  Our introduction does not intend to give an exhausting review on the pharmacological and traditional knowledge of Z. multiflora. The outline (as explained above) was identification of semi-polar metabolite of Z. multiflora and their antifungal activity.

  1. Results: Page 2, Line number 81, 82: What’s the different between geographically and environmentally diverse??

A: All information of geographic and climatic conditions of sampled plants in natural habitats were explained in Table S4 (Supplementary materials).

  1. Table 2: Serial no 43,44,45: Author could have written it as Zataroside A and Zataroside B as well.

A: We listed compounds 43 and 44 as Zataroside- isomer#1 and #2, because NMR analysis are needed to identify A and B isomer.

  1. Table 2: Serial no 48-55: What about all this diterpenoids?? Will they be identified or will be left??

A: Yes, the diterpenoids will be identified using NMR in future work, now added to the outlook.

  1. Page 7, Line number 178: Where is the reference of authentic reference compounds??

A: All information of authentic reference compounds were given in Table S5 (Supplementary materials).

  1. Discussion: Page 12, Line number 380-385: Simplify the statement. It seems to be little confusing. A: Simplified.
  1. Page 12, Line number 389-390: These are classes of compounds. Did the author found similarity among the species in their respective compounds also. Need clarifications in this aspect. If they have found similar compounds then mention the names.

A: Yes, we found similarity among the populations and explained it in results section 2.4. (Lines 300-312).

  1. Page 12, Line number 425-426: Where are the references for well documented antifungal activity of flavonoids?? Need to cite reference and need to discuss.

A: The following lines (lines 423-432) documented antifungal activity of flavonoids.

  1. Page 13, Line number 428-435: Probable mechanism of action needs to be provided herewith the discussion section for antifungal activity.

A: The information is provided (lines 502-503).

  1. Experimental: Page 14, Line number 527-528: Where is the latitude, longitude, altitude of the collection sites??

A: All information of geographic and climatic conditions of sampled plants in natural habitats are given in Table S4 (Supplementary materials).

  1. Page 14, Line number 529-531: Who exactly did the identification of the plant species used for the research??

A: the plant was identified by Prof. Javad Hadian and a voucher specimen (No. MPH-1799) was deposited in the Herbarium of Medicinal Plants and Drugs Research Institute, Shahid Beheshti University, Iran (Lines 526-528).

  1. Page 15, Line number 567-568: Why did not the author used incubator??

A: We used the incubator for antifungal assays. Plates were incubated at 20 °C in incubator.

  1. Page 15, Line number 569-572: Write the formula in their desired form. The used formula and for the assay as well author need to cite one more valid reference. A: Information provided.
  1. Additionally, the language of the manuscript needs to be improved. There are some grammatical and typical mistakes are there. The manuscript needs to be revised by a language expert.

A: The manuscript has been checked for language again.

Round 2

Reviewer 2 Report

The article can be accepted.

Reviewer 5 Report

Dear Authors

All the suggestions  given by me has been resolved properly Now the MS is ready for publication